# *SCGB1D2* inhibits growth of *Borrelia burgdorferi* and affects susceptibility to Lyme disease

Satu Strausz[1,2,3,4], Erik Abner [5,15], Grace Blacker[6,15], Sarah Galloway[6,15], Paige Hansen [6,7,15], Qingying Feng[7], Brandon T. Lee[6,7], Samuel E. Jones [1], Hele Haapaniemi[1], Sten Raak[5], George Ronald Nahass[6,7,8], Erin Sanders[6,7], FinnGen*, Estonian Genome Centre*, Pilleriin Soodla[9], Urmo Võsa [5], Tõnu Esko[5], Nasa Sinnott-Armstrong [1,2,10], Irving L. Weissman [6], Mark Daly[1], Tuomas Aivelo [11], Michal Caspi Tal [6,7] ✉ & Hanna M. Ollila[1,12,13,14] ✉

Lyme disease is a tick-borne disease caused by bacteria of the genus *Borrelia*. The host factors that modulate susceptibility for Lyme disease have remained mostly unknown. Using epidemiological and genetic data from FinnGen and Estonian Biobank, we identify two previously known variants and an unknown common missense variant at the gene encoding for Secretoglobin family 1D member 2 (*SCGB1D2*) protein that increases the susceptibility for Lyme disease. Using live *Borrelia burgdorferi (Bb)* we find that recombinant reference SCGB1D2 protein inhibits the growth of *Bb* in vitro more efficiently than the recombinant protein with SCGB1D2 P53L deleterious missense variant. Finally, using an in vivo murine infection model we show that recombinant SCGB1D2 prevents infection by *Borrelia* in vivo. Together, these data suggest that *SCGB1D2* is a host defense factor present in the skin, sweat, and other secretions which protects against *Bb* infection and opens an exciting therapeutic avenue for Lyme disease.

Lyme disease (i.e., borreliosis) is an infectious disease caused by bacteria of the genus *Borrelia* and transmitted by ticks. While most individuals respond to standard antibiotics, others develop severe infection which requires intensive antibiotic treatment and may result in chronic illness[1,2]. Seasonality of the disease is well characterized in the Northern hemisphere, and an increasing number of patients with Lyme disease have emerged during recent decades[3,4]. Yet the biological risk factors and disease mechanisms for infection or severe illness are still only partially understood. Therefore, we performed a genome-wide association study (GWAS) on Lyme disease assessed by the

[1]Institute for Molecular Medicine Finland, Helsinki Institute of Life Science, University of Helsinki, Helsinki, Finland. [2]Department of Genetics, Stanford University School of Medicine, Stanford, CA, USA. [3]Department of Oral and Maxillofacial Surgery, Helsinki University Hospital and University of Helsinki, Helsinki, Finland. [4]Department of Plastic Surgery, Cleft Palate and Craniofacial Center, Helsinki University Hospital and University of Helsinki, Helsinki, Finland. [5]Estonian Genome Centre, Institute of Genomics, University of Tartu, Tartu, Estonia. [6]Institute for Stem Cell Biology & Regenerative Medicine, Stanford University School of Medicine, Stanford, CA, USA. [7]Department of Biological Engineering, Massachusetts Institute of Technology, Cambridge, MA, USA. [8]Richard and Loan Hill Department of Biomedical Engineering, University of Illinois at Chicago, Chicago, IL, USA. [9]Department of Infectious Diseases, Internal Medicine Clinic, Tartu University Hospital, Tartu, Estonia. [10]Herbold Computational Biology Program, Fred Hutchinson Cancer Center, Seattle, WA, USA. [11]Organismal and Evolutionary Biology Research Program, University of Helsinki, Helsinki, Finland. [12]Broad Institute of MIT and Harvard, Cambridge, Massachusetts, MA, USA. [13]Center for Genomic Medicine, Massachusetts General Hospital, Boston, MA, USA. [14]Anesthesia, Critical Care, and Pain Medicine, Massachusetts General Hospital and Harvard Medical School, Boston, MA, USA. [15]These authors contributed equally: Erik Abner, Grace Blacker, Sarah Galloway, Paige Hansen. *Lists of authors and their affiliations appear at the end of the paper. ✉e-mail: mtal@mit.edu; hanna.m.ollila@helsinki.fi

International Classification of Diseases (ICD)−9 and 10 codes and explored phenotypic and genetic risk factors with the goal of fine-mapping the most significant genetic associations and understanding the underlying biology.

We utilized data from 617,731 individuals, including 25,355 individuals with Lyme disease who have participated in the FinnGen project or in the Estonian Biobank to estimate the effect of genetic variation on Lyme disease. The diagnoses were derived from ICD-codes in the Finnish and Estonian national hospital and primary care registries. In FinnGen 7354 (1.8%) and in Estonian Biobank 18,001 participants (8.8%) had received diagnosis for Lyme disease (Supplementary Table 1).

## Results

### Human leukocyte antigen and Toll-like receptor 1 loci affect susceptibility to Lyme disease

We identified three genetic loci in a meta-analysis combining two GWAS studies of Lyme disease ($P < 5.0 \times 10^{-8}$, Fig. 1 and Supplementary

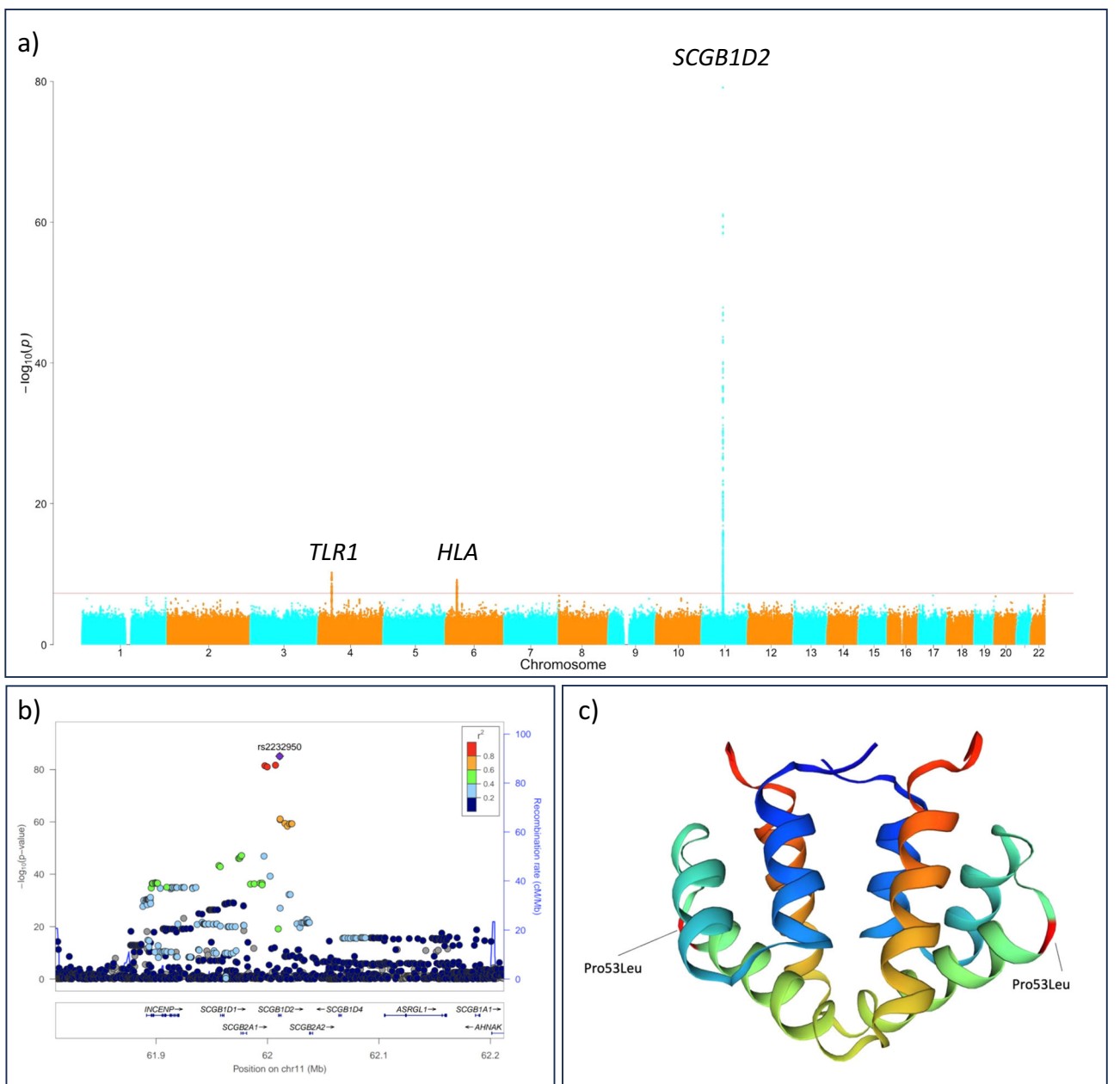

**Fig. 1 | Overview of genetic landscape of Lyme disease risk factors. a** Manhattan plot for the genome-wide association study (GWAS) of Lyme disease (LD) including 25,355 LD cases and 592,376 controls with Firth logistic regression model. All tests were two-sided. For each genetic variant, the x-axis shows chromosomal position, while y-axis shows the Bonferroni-adjusted $-\log_{10}(P)$ -value. The horizontal line indicates the genome-wide significance threshold of $P = 5.0 \times 10^{-8}$. Three genetic loci were identified at a genome-wide significance level: *SCGB1D2*, *HLA-DQB1*, and *TLR1*. **b** Locus Zoom plot shows associated Bonferroni-adjusted *P* values on the $\log_{10}$ scale on the vertical axis, and the chromosomal position along the horizontal axis. Purple diamond indicates single nucleotide polymorphism (SNP) at a locus with the strongest associated evidence. Linkage disequilibrium (LD, $r^2$ values) between the lead SNP and the other SNPs are indicated by color. **c** Schematic illustration for the protein structure of SCGB1D2 where a missense variant rs2232950 is causing an amino acid substitution from proline (Pro) to leucine (Leu). The structure is alpha-helical and forms an antiparallel dimer of the two monomers. There is a cavity which can accommodate small to medium-sized ligands like steroids and phospholipids between the two dimers.

Table 2). One of these was, perhaps unsurprisingly, located at the human leukocyte antigen (HLA) region on chromosome 6. The role of HLA is well-established to affect predisposition to infectious and autoimmune traits[5,6], and this association highlights the overall importance of the HLA in Lyme disease. Furthermore, the lead variant (rs9276610, $P = 6.56 \times 10^{-10}$) is located at the HLA class II locus. In addition, both FinnGen and Estonian biobank showed a significant association between HLA class II locus and Lyme disease with HLA-DQB1 being the closest gene in both cohorts (Supplementary Fig. 1).

To understand the regional association at the HLA locus, we utilized a population-specific high-resolution HLA imputation panel from FinnGen to estimate associations at the variant level, at the HLA allele level, and finally at the amino acid level, and fine-mapping associations from the HLA class I and class II loci. The meta-analysis lead variant, rs9276610, was in partial LD *HLA-DQB1*06:02* ($r^2 = 0.558$). We postulated that population-specific differences may exist with infectious traits where exposure and allele frequency both affect the frequency of single alleles, and therefore tested associations between cohorts separately. The lead variant from FinnGen (rs9273375) was in high LD with *HLA-DQB1*06:02* ($r^2 = 0.901$, Supplementary Table 3) and consequently we saw the strongest association for the FinnGen lead variant with *HLA-DQB1*06:02* followed by *DRB1*15:01* and *DRB5*01:01* (Supplementary Table 4) which are part of the same HLA haplotype (dominant model's $P = 4.4 \times 10^{-11}$, Supplementary Table 5). Similarly, when adjusting the analysis for *HLA-DQB1*06:02*, we discovered that there was only a weak residual signal remaining (rs9276610 $P$ conditioned = 0.02) at the HLA locus suggesting that *HLA-DQB1*06:02* explained the majority of the signal in this analysis, although larger data sets might clarify the possible additional signals at the HLA locus. In contrast, the lead variant from Estonian Biobank (rs28371212), which is located at the HLA class II locus, was not in high LD with any HLA allele or amino acid (Supplementary Table 6).

The HLA molecules present foreign peptides such as those from viruses or bacteria to T lymphocytes. Allelic diversity at the HLA genes differentiate which peptides can be presented and presentation is modified by individual amino acids unique for the HLA allele. Furthermore, amino acids can be highly specific for just one allele or sometimes shared across several HLA alleles. We therefore examined the preferential association across amino acid level variation from the HLA alleles. In FinnGen, we detected an association with amino acid Serine at DRB1 position 0, an amino acid specific for the DRB1*15:01 allele (beta = 0.17, se = 0.026, $P = 3.7 \times 10^{-11}$) and with DQB1 phenylalanine at position 9 that is found in HLA-DQB1*06:02 and in HLA-DQB1*04 alleles. These amino acids associations further suggest the *HLA-DRB1*15:01-HLA-DQB1*06:02* haplotype as a susceptibility haplotype for Lyme disease (Supplementary Table 7). The specific alleles, and especially the *HLA-DQB1*06:02* allele or alleles in high linkage disequilibrium (LD) such as *HLA-DRB1*15:01*, have been previously associated with influenza-A infection[7,8], autoimmune diseases such as multiple sclerosis[9], and with type-1 narcolepsy[10].

Finally, the different lead variants between FinnGen and Estonian Biobank with significant SNP level association in both raise the possibility of two independent signals, uncaptured allelic diversity, or regulatory variants at the HLA locus. Therefore, we conditioned both cohorts with the lead signal (rs9276610). We did not observe evidence for additional regional associations at the locus (Supplementary Fig. 2) and such associations need to be examined in future studies and larger cohorts.

We also observed an association within the Toll-like receptor 1 (*TLR1*) locus (rs17616434, $P = 6.11 \times 10^{-11}$). TLR1 in chromosome 4 and, also TLR2, are interesting as coding variants at the loci have been previously associated with Lyme disease[11]. In our study, we observed an association with TLR1 locus and the lead variant is a non-coding variant at 5'UTR of *TLR1* and an eQTL for *TLR1, TLR6* and *TLR10* and associates robustly in FinnGen and in Estonian Biobank (Supplementary Fig. 3).

Additionally, it is in high LD with two missense variants at *TLR1* (rs5743618, Ser602Ile, $r^2 = 0.90$; and rs4833095, Asn248Ser, $r^2 = 0.89$). The *TLR1* locus has been reported as the main association for individual variation for TLR stimulation and response[12]. Furthermore, TLR1 SNPs, including the missense variants, have been associated with asthma or allergy and lymphocyte counts[13–15] and examined in infectious traits[16,17], including Lyme disease[11].

## Missense variants at *SCGB1D2* affect susceptibility to Lyme disease

While HLA and TLR associations provide a proof of principle establishing Lyme disease as an infectious trait that is modified by genetic risk factors, the strongest and most compelling association was found at Secretoglobin family 1D member 2 locus (*SCGB1D2*, rs2232950, $P = 8.10 \times 10^{-86}$, Fig. 1) and the same lead variant associating in FinnGen (*SCGB1D2*, rs2232950, $P = 6.12 \times 10^{-31}$) and in Estonian Biobank (*SCGB1D2*, rs2232950, $P = 1.02 \times 10^{-56}$, Supplementary Fig. 4). Fine-mapping the locus revealed rs2232950 as part of the credible set including six single nucleotide polymorphisms and as the most likely causal variant (posterior probability = 0.22) with no additional genome-wide significant signals after conditioning for the main effect (Supplementary Fig. 5).

Rs2232950 is a common missense variant with an alternative allele frequency of 40%. The variation marked by rs2232950 at position 158 C > T of *SCGB1D2* results in a substitution from proline to leucine (Pro53Leu) and predicted deleterious by SIFT[18] and Polyphen[19] algorithms. Furthermore, prolines are often the initiators of α-helix structures[20] and Pro53 is the first residue in SCGB1D2 H3-helix backbone. Therefore, a Pro>Leu substitution at this position will likely destabilize the α-helical structure, as the downstream amino acids (Val56 and Ala57) will lose an anchor-point (Supplementary Fig. 6). It could be hypothesized that such a loss in structural integrity leads to an altered SCGB1D2 dimerization affinity, since the H3-helices of secretoglobins interact with the H3-H4-helices of the corresponding dimeric partners[21].

To understand the possible function of the variant, we examined its association in phenome-wide analysis (PheWAS) across 2202 disease endpoints in FinnGen and publicly available GWAS data (Supplementary Fig. 7). In addition to Lyme disease, we observed an association with hospitalized spirochetal infections (Fig. 2). As spirochetal diseases include Lyme disease, we examined if the association was due to hospitalized Lyme disease patients and, consequently observed that the majority of hospitalized individuals had Lyme disease ($N = 2357$ of 2492 individuals with hospital-level spirochetal infection).

Furthermore, to elucidate the possible broader association of SCGB1D2 Pro53Leu with pathogens we analysed association across 36 different disease categories, including tick-mediated diseases (tick-borne encephalitis), general arthropod behavioral markers (scabies), spirochete bacterium phylum members (syphilis) and other bacterial diseases (e.g., sepsis, scarlet fever, and erysipelas) of which none were associated with Lyme disease SCGB1D2 Pro53Leu variant (Supplementary Fig. 8).

## *SCGB1D2* is expressed in the skin and secreted by sweat gland cells

*SCGB1D2* belongs to a Secretoglobin protein family in which all members, except for *SCGB1D2*, are found in upper airway tissues and are generally expressed by secretory tissues of barrier organs[22,23]. We used data from the Genotype-Tissue Expression (GTEx)[24] release 8 to understand the tissue distribution of *SCGB1D2*. GTEx contains RNA expression samples from 948 donors across 54 tissues. The highest expression of *SCGB1D2* occurred in two skin tissues, sun-exposed and not sun-exposed skin (transcripts per million (TPM) = 258.2 and TPM = 163.2, respectively, Supplementary Fig. 9). Expression was also

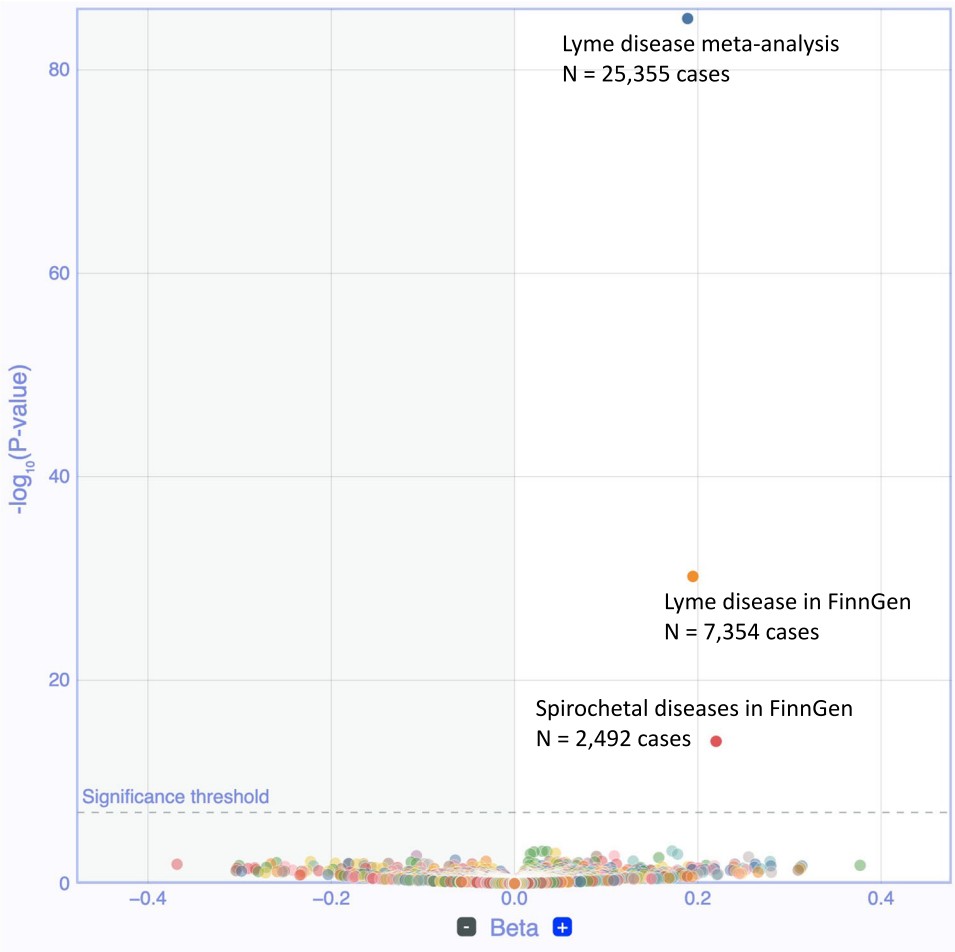

**Fig. 2 | Phenome-wide associations of *Secretoglobin 1D* across disease traits.** Volcano plot of Phenome-wide associations (PheWAS) from rs2232950 and 2,202 disease endpoints from FinnGen. Each point represents a trait. The vertical axis presents associated Bonferroni-adjusted *P* values at −log10 scale and the horizontal axis shows beta values. All tests were two-sided. Other spirochetal diseases contain individuals who have been treated at hospital inpatient or outpatient clinics and is partially overlapping with Lyme disease as spirochetal diseases include individuals with Lyme disease.

observed in other secretory tissues including salivary gland, mammary tissue, and the uterus.

As skin is the first tissue for *Bb* exposure, we were interested in understanding the cell type distribution of *SCGB1D2* expression in the skin. Indeed, earlier studies have suggested that *SCGB1D2* expression maybe even specific to sweat glands in the skin[25,26]. Therefore, we visualized previously published single-cell sequencing data across cell types that are observed in the skin: fibroblasts, keratinocytes, lymphatic endothelial cells, melanocytes, pericytes, smooth muscle cells, sweat gland cells, T cells, and vascular endothelial cells[25]. The visualization showed that *SCGB1D2* expression was specific to the sweat gland cells (Supplementary Fig. 10), suggesting that SCGB1D2 may be secreted on the skin as part of sweat. Antimicrobial peptides such as dermcidin, lysozyme, lactoferrin, psoriasin, cathelicidin, and β-defensins have been previously discovered in human sweat[27], and raises the possibility that SCGB1D2 may have antimicrobial properties.

### SCGB1D2 inhibits growth of *B. burgdorferi*

To investigate the impact of SCGB1D2 protein encoded by the reference genotype on *Bb* growth, we performed a *Bb* growth assay using recombinant reference SCGB1D2. We discovered that SCGB1D2 inhibited the growth of *Bb* at 24 h (ANOVA F (4, 18) = 5.77, P = 0.0036) with significant growth inhibition at 8 and 16 µg/mL concentrations (P = 0.0091, P = 0.0025 respectively, Fig. 3a). The growth inhibition

was also significant at 72 h (ANOVA F (4, 18) = 7.67, P = 0.0009) time point at 8 and 16 µg/mL concentrations (P = 0.0137, P = 0.0010, respectively, Fig. 3b). Furthermore, despite an increase in the *Bb* count over time the growth inhibition was still notable at 72 h suggesting that the effect is not transient (Fig. 3a–c). Furthermore, the growth inhibition was dose-dependent at various SCGB1D2 concentrations over time (Fig. 3c).

We then tested whether the SCGB1D2 P53L amino acid substitution encoded by the variant genotype might affect the SCGB1D2 protein function or even have a different impact on *Bb* growth compared to the reference SCGB1D2 protein without P53L substitution. To test this, we performed a *Bb* growth assay using recombinant reference SCGB1D2 compared to recombinant variant SCGB1D2 P53L. We discovered that approximately twice the amount of the variant SCGB1D2 P53L was needed to achieve similar inhibition to SCGB1D2 and that the variant SCGB1D2 P53L inhibited *Bb* growth only at the highest concentration of 16 µg/mL (Fig. 3d and Supplementary Fig. 11). Similarly, while the reference SCGB1D2 overall inhibited *Bb* growth to a greater extent than SCGB1D2 P53L variant at 8 mg/mL, (P = 0.046, Fig. 3) both reference SCGB1D2 and SCGB1D2 P53L variant were able to inhibit *Bb* growth at the highest concentration where the effect was possibly saturated (P = 0.119, Fig. 3d, e).

We observed that in the presence of 16 µg/mL of variant SCGB1D2 P53L, and 8 or 16 µg/mL of reference SCGB1D2, the *Bb* count reduces slightly from the starting amount to a trough around the 24-h time

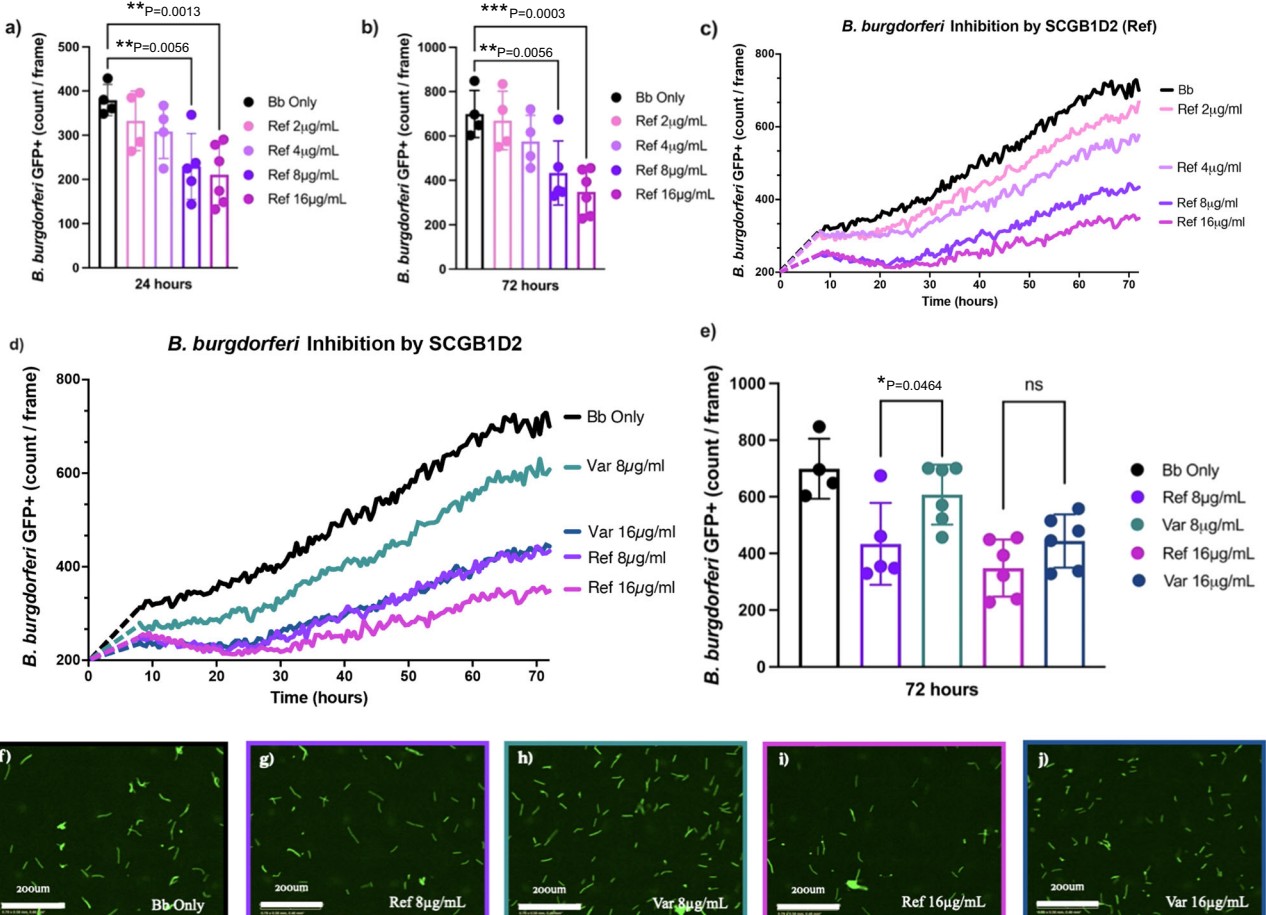

**Fig. 3 | Functional analyses of SCGB1D2 protein and *Borrelia burgdorferi*.** *Borrelia burgdorferi* (*Bb*) was incubated alone (black), or with either reference (Ref) or variant (Var) SCGB1D2 recombinant protein at 2, 4, 8, or 16 µg/mL. IncuCyte analysis showing the count of green fluorescent protein (GFP)-expressing *Bb* per image with reference SCGB1D2 protein at 24 h time point (**a**) and at 72 h time point (**b**). Count of GFP-expressing *Bb* spirochetes per image over time with all concentrations of reference SCGB1D2 protein (**c**). Count of GFP-expressing *Bb* spirochetes per image over time with SCGB1D2 reference and SCGB1D2 P53L recombinant proteins at 8 and 16 µg/mL concentration (**d**). Comparison of reference SCBG1D2 and variant SCGB1D2 P53L at 8 and 16 µg/mL concentrations over time, and **e** at 72 h. *n* = 4–6 technical replicates per condition, each technical replicate is a single well's mean count/frame value from 9 IncuCyte fields of view for figures **a**, **b**, and **e**. *Bb* replicate rapidly and significant heterogeneity in plasmid retention can emerge over time, therefore in this case, it's essential to compare technical replicates. Data are presented as mean values ± SD and were analyzed via one-way ANOVA. Dotted lines in (**c**, **d**) connect from the amount of bacteria plated in each well until imaging began 8 h later. Representative IncuCyte images for each treatment condition at 140 h, scale bars at 200 µm (**f–j**). ***$P = 0.001$, **$P < 0.01$, *$P < 0.05$ and ns not significant.

point (Fig. 3d). This may be indicative of bacterial killing by SCGB1D2, especially since this graphical trough is not observed in the *Bb* only condition or with 8 µg/mL of variant SCGB1D2 P53L.

In order to label dead cells and differentiate bacterial death from growth inhibition, propidium iodide (PI) was included in a repeat experiment of *Bb* growth inhibition. We observed the presence of dead spirochetes in the assay, and the cytotoxicity of prolonged PI exposure. Furthermore, a significant proportion of *Bb* were bound to PI compared to *Bb* conditions without PI, even in the absence of SCGB1D2 proteins (Supplementary Figs. 12a, 13). To account for the increased PI binding, we examined the proportion of PI bound to *Bb* in the presence of either SCGB1D2 reference or variant protein but did not observe a significant difference in *Bb* death at 24 h compared to *Bb* with PI control (Supplementary Fig. 12b). Nevertheless, the graphical trends of *Bb* death in the presence of SCGB1D2 protein were similar to observations of growth inhibition (Fig. 3). Specifically, 16 µg/mL of reference SCGB1D2, which resulted in the most growth inhibition, also resulted in most PI binding ($P = 0.0822$). Further investigation is required to determine the SCGB1D2 mechanism of action against *Bb* and to differentiate bacterial killing from growth inhibition. Representative

images of *Bb* from each treatment condition after 140 h are also shown (Fig. 3f–j).

## SCGB1D2 prophylaxis prevents intradermal infection with Bb in vivo

To investigate the impact of SCGB1D2 on *Bb* establishing infection in vivo, we injected C57BL/6 J female mice aged at 10 weeks at the time of infection intradermally with *Bb* incubated with either SCGB1D2 or SCGB3A1, a secretoglobin that does not associate with Lyme disease. Furthermore, we used two different *Bb* strains genetically modified to express luciferase: N40D10/E9 and ML23[28,29]. All intradermal injections of *Bb* were verified on day 0 (Fig. 4a, c) by confirming a luciferase signal with an In Vivo Imaging System (IVIS) (Perkin Elmer). The luciferin signals were quantified by gating individual total flux (photons/seconds) on days 0, 3, and 10 (Fig. 4b, d). Pre-incubation of either ML23 or N40D10/E9 strains of *Bb* with SCGB1D2 protein prevented *Bb* from establishing infection (Fig. 4a, c). We observed a significant difference in the total flux of ML23 (pairwise Wilcoxon test, $P = 0.024$) and N40D10/E9 (pairwise Wilcoxon test, $P = 0.024$) *Bb* bacterial load via imaging starting

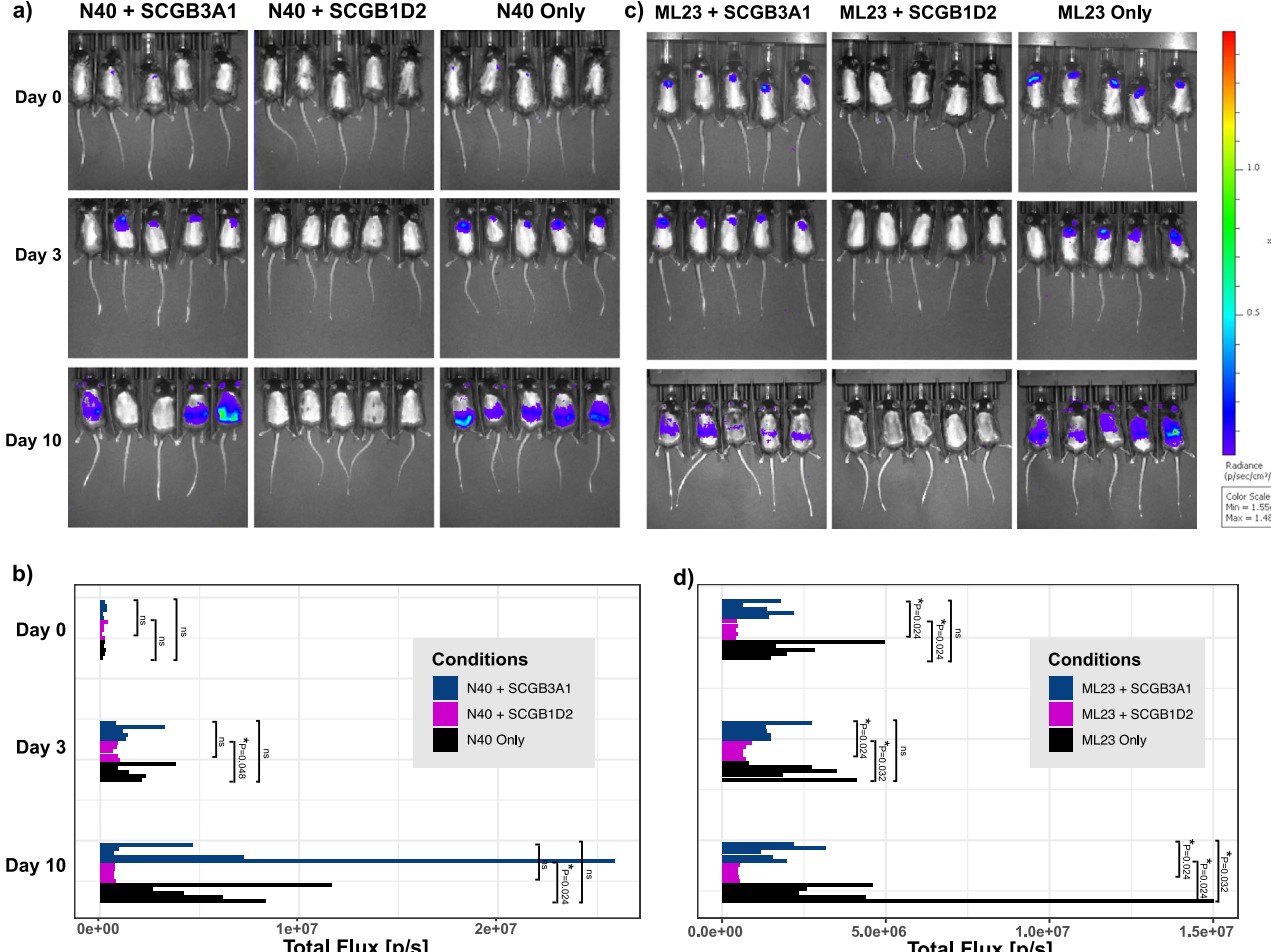

**Fig. 4 | In vivo imaging system (IVIS) quantification of SCGB1D2 prophylactic effect on intradermal infection with *Bb*.** Dissemination of two *Bb* luciferase-containing strains, N40D10/E9 and ML23, was quantified by gated signals from individual mice 15 min after being injected with 277 mg/kg sterile filtered D-luciferin dissolved in phosphate-buffered saline (PBS). **a** IVIS imaging of 1 min exposures of mice at days 0, 3, and 10 post-infection of *Bb* (N40D10/E9) where the *Bb* had been co-incubated with SCGB1D2 or SCGB3A1 or no protein control. Quantified total flux (p/s) of each mouse (left to right) is shown in (**b**) in sequential order (top to bottom). All images were normalized across all conditions and all time points. **c** IVIS imaging of mice at days 0, 3, and 10 post-infection of *Bb* (ML23) co-infected with SCGB1D2 or SCGB3A1, where quantified total flux (p/s) of each mouse (left to right) is shown in (**d**) in sequential order (top to bottom). All images were normalized across all conditions and all time points. The mean from each treatment condition is compared against the means of all other treatment conditions in pairwise Wilcoxon test. *$P < 0.05$ and ns not significant.

from day 3 onwards. Gating on just the site of infection also reveals a significant difference in total flux on the day of infection with N40D10/E9 (Supplementary Fig. 14, two-way ANOVA $F_{(2, 12)} = 6.691$, $P = 0.0112$). Through day 10 post-infection, we do not observe evidence of infection or dissemination of *Bb*, which had been prophylactically incubated and co-injected with SCGB1D2 (Fig. 4a, c).

In both N40D10/E9 and ML23 *Bb*, which was pre-incubated and co-injected with SCGB3A1, showed successful infection in mice. We observed significant differences between ML23 pre-incubated and co-injected with SCGB3A1 and SCGB1D2 (Wilcoxon signed-rank test, $P = 0.000183$).

## Discussion

In this study, we report genetic variants that affect Lyme disease susceptibility. Most notably, we identified an association with a deleterious missense variant in the *SCGB1D2* gene. Secretoglobin protein family members have previously been described to function at the epithelial barrier and participate in innate immune pathways within airways[30]; however, the function of SCGB1D2 has been unknown. It is noteworthy that this *SCGB1D2* P53L variant appears

quite specific for Lyme disease and has not been previously reported as associated with any other disease, phenotype, or infection. Using expression and single-cell analysis, we observe that *SCBG1D2* has the highest expression in the skin and sweat gland cells. Furthermore, we characterized the function of SCGB1D2 protein both in vitro and in vivo. We show that recombinant SCGB1D2 significantly inhibits *Bb* growth in vitro, and that around twice as much SCGB1D2 variant protein is required to achieve the same level of *Bb* growth inhibition as reference SCGB1D2. In addition, we show through in vivo imaging of luciferase-containing *Bb* in the mouse body that when *Bb* is incubated and co-injected with recombinant SCGB1D2 protein, it significantly inhibits *Bb* infection kinetics compared to *Bb* only *or Bb* incubated and co-injected with recombinant SCGB3A1, another secretoglobin that does not associate with Lyme disease. These findings suggest that *SCGB1D2* is a risk locus for Lyme disease and suggests that SCGB1D2 may act by restricting *Bb* growth. Overall, our results elucidate a mechanism by which secretoglobin may provide protection against *Bb* infection in the skin. In the scope of infectious diseases, such disease-specific associations help teach us about host defenses that impact Lyme disease.

In the context of immune and infectious traits, it is nearly impossible to discuss the risk for infection outside the scope of HLA. Also, in our study, we identified an association between Lyme disease and the HLA class II locus with a likely allelic risk from *HLA-DQB1\*06:02*. Similarly, we identified an association with the *TLR1* locus, which has been previously studied in the context of Lyme disease[11]. These findings highlight the overall impact that HLA alleles have on infectious and autoimmune traits and raises two interesting points. First, the balance between innate and adaptive immune responses in clearing *Bb* infection in the human host remains an active topic of research. HLA class II alleles such as *DQB1\*06:02* are critical in modulating adaptive immune responses, whereas Toll-like receptors constitute key regulators of the innate immune arm. Our findings suggest that both innate and adaptive immune responses, through Toll-like receptors for innate immunity and primarily T-cell or B-cell mediated immunity for adaptive immunity, have an important role in Lyme disease. Second, both *HLA-DRB1\*15:01* and *DQB1\*06:02* have been previously implicated in brain autoimmune and infectious diseases such as multiple sclerosis, narcolepsy, and influenza-A[7,9,10]. *HLA-DRB1* has additionally been implicated in Lyme arthritis, and specifically the *HLA-DRB1\*15:01* allele was found to be more common in antibiotic-refractory Lyme arthritis patients than antibiotic-responsive patients[2]. Furthermore, selection works relatively strongly at the HLA locus as infectious challenges can rapidly favor an HLA allele that protects the population against a particular infection. In addition, the frequency of HLA alleles varies between populations and can affect the power to detect associations in different populations. Consequently, studies in Lyme disease will benefit from even larger genetic studies. Our findings provide yet another infectious disease trait that is associated with these same HLA alleles and raises the possibility that the same variants that contribute to infectious diseases also affect autoimmune and chronic disease traits in general.

Finally, our findings provide a compelling association with the *SCGB1D2* locus and P53L variant in particular. This study suggests that the SCGB1D2 protein may inhibit bacterial growth. Additionally, the results indicate SCGB1D2 as a host defense mechanism against *Borrelia* infection and against Lyme disease. This finding provides a biological mechanism to explore as a therapeutic avenue for drug development to prevent and treat Lyme disease.

## Methods
Our research complies with ethical guidelines and regulations. We provide detailed information in the Inclusion and ethics section of the manuscript.

### FinnGen
FinnGen (www.finngen.fi/en) is a joint research project of the public and private sectors, launched in Finland in the autumn of 2017, that aims to genotype 500,000 Finns, including prospective and retrospective epidemiological and disease-based cohorts as well as hospital biobank samples. FinnGen combines genome data with longitudinal health care registries using unique personal identification codes allowing data collection and follow-up even over the whole life span[31].

The FinnGen data release 10 is composed of 412,181 Finnish participants. The diagnosis of Lyme disease was based on ICD-codes (ICD −10: A69.2, ICD-9: 1048 A), which were obtained from the Finnish national hospital and primary care registries, including 7354 individuals with Lyme disease and 404,827 controls (Supplementary Table 1).

### Estonian Biobank
The Estonian Biobank is a population-based biobank of the Estonian Genome Center at the University of Tartu. Its cohort size is 212,955 participants, which closely reflects the age, sex, and geographical distribution of the Estonian population. Lyme disease was based on

ICD-10 code A69.2. The full analysis included 18,001 cases and 187,549 disease-free controls with both genome-wide genotyping and electronic health record data information (Supplementary Table 1).

### Genotyping and imputation
FinnGen samples were genotyped with Illumina and Affymetrix chip arrays (Illumina Inc., San Diego, and Thermo Fisher Scientific, Santa Clara, CA, USA). Genotype calls were made with GenCall and zCall algorithms for Illumina and the AxiomGT1 algorithm for Affymetrix data. Chip genotyping data produced with previous chip platforms and reference genome builds were lifted over to build version 38 (GRCh38/hg38) following the protocol described here: dx.doi.org/10.17504/protocols.io.nqtddwn.

In sample-wise quality control, individuals with ambiguous gender, high genotype missingness (>5%), excess heterozygosity (±4 standard deviation), and non-Finnish ancestry were excluded. In variant-wise quality control variants with high missingness (>2%), low Hardy–Weinberg equilibrium *P* value ($<1.0 \times 10^{-6}$) and minor allele count <3 were excluded. Prior imputation, chip genotyped samples were pre-phased with Eagle 2.3.5[32] with the default parameters, except the number of conditioning haplotypes was set to 20,000.

Genotype imputation was done with the population-specific SISu v4 reference panel. The variant call set was produced with the Genomic analyses toolkit (GATK) HaplotypeCaller algorithm by following GATK best-practices for variant calling. Genotype-, sample- and variant-wise quality control was applied in an iterative manner by using the Hail framework v0.1 and the resulting high-quality whole genome sequenced data for 3775 individuals were phased with Eagle 2.3.5. Post-imputation quality control involved excluding variants with INFO score <0.7.

All EstBB participants have been genotyped at the Core Genotyping Lab of the Institute of Genomics, University of Tartu, using Illumina Global Screening Array v3.0_EST. Samples were genotyped and PLINK format files were created using Illumina GenomeStudio v2.0.4. Individuals were excluded from the analysis if their call-rate was <95% or if sex defined based on heterozygosity of the X chromosome did not match the sex in phenotype data. Before imputation, variants were filtered by call-rate <95%, HWE *p* value ($<1 \times 10^{-4}$) (autosomal variants only), and minor allele frequency <1%. Variant positions were in build 37 and were lifted over to build 38 using Picard. Phasing was performed using the Beagle v5.4 software[33]. Imputation was performed with Beagle v5.4 software (beagle.22-Jul22.46e.jar) and default settings. The dataset was split into batches of 5000. A population-specific reference panel consisting of 2695 WGS samples was utilized for imputation and standard Beagle hg38 recombination maps were used. Based on the principal component analysis, samples who were not of European ancestry and samples who were twins or duplicates of included samples were removed.

### Statistical methods
**FinnGen.** We performed genome-wide association testing as implemented in the Regenie[34] v2.2.4 using FinnGen Regenie pipelines (https://github.com/FINNGEN/regenie-pipelines). Analysis was adjusted for current age or the age at death, sex, genotyping chip, genetic relationship, and first ten principal components. To examine the *SCGB1D2* locus and its genomic variation's causality to Lyme disease in more detail, we fine-mapped this region utilizing the "Sum of Single Effects"−model, called *SuSiE*[35].

**EstBB.** We analyzed the data using Regenie[34] accounting for sex, age, and population structure by genetic relatedness matrix. Association analysis in Estonian Biobank was carried out for all variants with an INFO score >0.4 using the additive model as implemented in Regenie v2.2.4 with standard binary trait settings[34]. Logistic regression was

carried out with adjustment for current age, age2, sex, and ten PCs as covariates, analyzing only variants with a minimum minor allele count of 2.

**Meta-analysis.** We computed a fixed-effects meta-analysis using summary statistics from the GWAS from FinnGen and the Estonian Biobank. Analysis was run with METAL[36].

### PheWAS

We performed a phenome-wide association analysis in FinnGen by retrieving association statistics for rs2232950 and all core phenotypes from FinnGen. We considered $P$ values significant if they passed Bonferroni correction for 2202 tests corresponding to $P$ value $5.2 \times 10^{-5}$.

Phenome-wide association analysis for 36 different traits was carried out using Regenie v2.2.4[34]. Logistic regression for a single variant (rs2232950) was carried out with adjustment for current age, age[2], sex, and ten PCs as covariates.

### Functional follow-up

We examined RNA expression across tissue types using GTEx[24] v8 using the fully processed and normalized gene expression matrixes for each tissue. These same values are used for eQTL calculations by GTEx.

We used previously published single-cell sequencing data[25] across cell types that are observed in the skin: fibroblasts, keratinocytes, lymphatic endothelial cells, melanocytes, pericytes and smooth muscle cells, sweat gland cells, T cells, and vascular endothelial cells, and tested *SCGB1D2* expression in these cell types (Supplementary Fig. 10).

### *B. burgdorferi* in vitro growth inhibition assay

Data for functional analysis of *Bb* was collected by the IncuCyte® S3 and analyzed in GraphPad Prism v9.2.0. We conducted ANOVA to explore the differences between groups, followed by Dunnett's post hoc test to determine whether the tested groups were significantly different from the control group. We used independent *T*-tests in comparisons where there were only two groups. In addition, we used a pairwise Wilcoxon test to compare each treatment condition against of all other treatments.

B31A3-GFP (Green Fluorescent Protein) *Bb* was cultured at 37 °C in Barbour–Stonner–Kelly with 4-(2-hydroxyethyl)-1-piper-azineethanesulfonic acid (HEPES) buffer media (BSK-H) complete with 6% rabbit serum (Millipore Sigma) and 1% Amphotericin B (Sigma-Aldrich). Bacterial concentration was determined by flow cytometry (Becton Dickinson LSRFortessa) such that 150,000 spirochetes per well of an ImageLock 96-well plate (Sartorius) were incubated with either 8 or 16 µg/mL of variant (TP607036;MKLSVCLLLVTLALCCYQ ANAEFCPALVSELLDFFFISEPLFKLSLAKFDAPLEAVAAKLGVKRCTDQMS LQKRSLIAEVLVKILKKCSV) or reference (TP607035; MKLSVCLLLVTL ALCCYQANAEFCPALVSELLDFFFISEPLFKLSLAKFDAPPEAVAAKLGVKR CTDQMSLQKRSLIAEVLVKILKKCSV) SCGB1D2 recombinant protein (Origene Technologies) in a total well volume of 150 uL. Incubation at 37 °C and 5% $CO_2$ occurred inside of an IncuCyte® S3 (Sartorius) to measure real-time fluorescent intensity and capture phase images over time. Images were acquired using a 20x objective at 300-ms exposure per field of view. In order to limit false positive background fluorescence, threshold values to determine GFP[+] events were set such that only *Bb* in phase and overlapped with green fluorescence events were counted.

A repeat *Bb* growth inhibition assay was performed as previously described, with the addition of 1.5 µL propidium iodide (Millipore Sigma) per well prior to incubation. After 24 h of incubation, samples were fixed in 4% paraformaldehyde, resuspended in flow cytometry buffer (2% FBS, 1 mmol EDTA, in phosphate-buffered saline (PBS)), and analyzed by flow cytometry.

### Murine *B. burgdorferi*-luciferase infection bioluminescent quantification with SCGB1D2 & SCGB3A1

N40D10/E9 (kindly gifted by Dr. Nikhat Parveen) and ML23 (kindly gifted by Dr. Jenny A. Hyde) luciferase *Bb* strains were cultured for 4 days at 37 °C in BSK-II media and complete with 6% rabbit serum (Millipore Sigma) at pH of 7.6. ML23 cultures had the addition of a selection antibiotic, kanamycin, at 300 µg/mL (Sigma-Aldrich). Bacterial concentration was determined by flow cytometry (Becton Dickinson LSR-II). 1,000,000 spirochetes per condition were resuspended in 500 µL of 0.2% uninfected B6 murine serum (diluted in PBS). Working with one strain at a time (N40D10/E9 and then ML23), spirochetes were incubated with either 20 µg/mL of SCGB3A1 recombinant protein (Aviva Systems Biology), 20 µg/mL of reference SCGB1D2 recombinant protein (Aviva Systems Biology), or no additional protein at 37 °C for 30 min. All recombinant proteins were provided to us by Aviva Systems Biology without any of the standardly added antimicrobials at our request. Investigators infecting the mice were blinded from the treatment conditions for the entire duration of the experiment and the analysis. Female C57BL/6 J mice ($n = 5$ per condition) aged 10 weeks were provided by Jackson Laboratories (Bar Harbor, ME) and were each injected intradermally with 50 µL of each *Bb* condition, such that each mouse was infected with a load of 100,000 *Bb*. Mice were anesthetized by isoflurane inhalation, then injected with the spirochetes intradermally. Cages of mice were infected in the following order: *Bb* + SCGB3A1, *Bb* + SCGB1D2, and *Bb* alone. To confirm infection, an injection of 100 µL sterile filtered D-luciferin reconstituted in sterile PBS at 277 mg/kg was done intraperitoneally. The hair on the backs of the mice was removed using an electric shaver, then the addition of Nair for 30 s to 1 min. Gauze wipes were used to clean the backs in the following order: soaked in 70% ethanol, soaked distilled water, and dry gauze. After 15 min, mice were arranged in the In Vivo Imaging System (Perkin Elmer) and then imaged for an exposure time of 1 min. This in vivo imaging using D-luciferin was later repeated at each of the indicated intervals (days 2, 3, 7,10).

### Analysis of in vivo imaging system (IVIS)

All downstream IVIS analyses were performed blinded in Living Image (Perkin Elmer). Individual total flux (p/s) values were achieved by rectangular gating of each mouse from the top of the nose cone to the tail tip. All images were normalized across all cages and time points with a set radiance range from $1.55e^4$ -$1.48e^6$ (p/sec/cm$^2$/sr) for visualization only. Downstream data analysis and visualization were done in R (version 4.1.1) with the following packages: ggpubr, ggplot2 and RColorBrewer.

### Inclusion and ethics

Patients and control subjects in FinnGen provided informed consent for biobank research, based on the Finnish Biobank Act. Alternatively, separate research cohorts, collected prior to the Finnish Biobank Act came into effect (in September 2013) and start of FinnGen (August 2017), were collected based on study-specific consents and later transferred to the Finnish biobanks after approval by Fimea (Finnish Medicines Agency), the National Supervisory Authority for Welfare and Health. Recruitment protocols followed the biobank protocols approved by Fimea. The Coordinating Ethics Committee of the Hospital District of Helsinki and Uusimaa (HUS) statement number for the FinnGen study is Nr HUS/990/2017.

The FinnGen study is approved by Finnish Institute for Health and Welfare (permit numbers: THL/2031/6.02.00/2017, THL/1101/5.05.00/ 2017, THL/341/6.02.00/2018, THL/2222/6.02.00/2018, THL/283/ 6.02.00/2019, THL/1721/5.05.00/2019, and THL/1524/5.05.00/2020), Digital and population data service agency (permit numbers: VRK43431/2017-3, VRK/6909/2018-3, VRK/4415/2019-3), the Social

Insurance Institution (permit numbers: KELA 58/522/2017, KELA 131/522/2018, KELA 70/522/2019, KELA 98/522/2019, KELA 134/522/2019, KELA 138/522/2019, KELA 2/522/2020, KELA 16/522/2020), Findata permit numbers THL/2364/14.02.2020, THL/4055/14.06.00/2020,THL/3433/14.06.00/2020, THL/4432/14.06/2020, THL/5189/14.06/2020, THL/5894/14.06.00/2020, THL/6619/14.06.00/2020, THL/209/14.06.00/2021, THL/688/14.06.00/2021, THL/1284/14.06.00/2021, THL/1965/14.06.00/2021, THL/5546/14.02.00/2020, THL/2658/14.06.00/2021, THL/4235/14.06.00/2021 and Statistics Finland (permit numbers: TK-53-1041-17 and TK/143/07.03.00/2020 (earlier TK-53-90-20) TK/1735/07.03.00/2021).

The Biobank Access Decisions for FinnGen samples and data utilized in FinnGen Data Freeze 8 include: THL Biobank BB2017_55, BB2017_111, BB2018_19, BB_2018_34, BB_2018_67, BB2018_71, BB2019_7, BB2019_8, BB2019_26, BB2020_1, Finnish Red Cross Blood Service Biobank 7.12.2017, Helsinki Biobank HUS/359/2017, Auria Biobank AB17-5154 and amendment #1 (August 17 2020), AB20-5926 and amendment #1 (April 23 2020), Biobank Borealis of Northern Finland_2017_1013, Biobank of Eastern Finland 1186/2018 and amendment 22 § /2020, Finnish Clinical Biobank Tampere MH0004 and amendments (21.02.2020 & 06.10.2020), Central Finland Biobank 1-2017, and Terveystalo Biobank STB 2018001.

The activities of the EstBB are regulated by the Human Genes Research Act, which was adopted in 2000 specifically for the operations of the EstBB. Individual-level data analysis in the EstBB was carried out under ethical approval 1.1-12/624 from the Estonian Committee on Bioethics and Human Research (Estonian Ministry of Social Affairs), using data according to release application 3-10/GI/31688 from the Estonian Biobank. All biobank participants have signed a broad informed consent form and information on ICD-codes is obtained via regular linking with the National Health Insurance Fund and other relevant databases, with a majority of the electronic health records having been collected since 2004[24].

Animal studies were performed at the Department of Comparative (DCM) at the Massachusetts Institute of Technology (MIT) (Cambridge, MA). All procedures and care guidelines were approved by the MIT Committee on Animal Care (CAC) (Protocol#1221-087-24).

## Reporting summary

Further information on research design is available in the Nature Portfolio Reporting Summary linked to this article.

## Data availability

Individual-level genotypes and register data from FinnGen participants can be accessed by approved researchers via the Fingenious portal (https://site.fingenious.fi/en/) hosted by the Finnish Biobank Cooperative FinBB (https://finbb.fi/en/). FinnGen summary statistics are available through https://www.finngen.fi/en.

The GWAS summary statistics data generated in this study have been deposited in the GWAS Catalog -database under accession code GCP000742.

## Code availability

Code to perform all analyses, including GWAS analyses, PheWAS plots, and fine-mapping of variants, is available at the FinnGen Github: https://github.com/FINNGEN/.

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

## Acknowledgements

We want to acknowledge the participants and investigators of the FinnGen study. We want to acknowledge the participants and investigators of the Estonian Biobank for their contribution. This work was carried out in part in the High-Performance Computing Center of the University of Tartu. We appreciate the ongoing support of the MIT Department of Comparative Medicine and the Preclinical Imaging Core for consultations and support during in vivo mouse experiments. This work has been supported by Emily and Malcolm Fairbairn Foundation (M.C.T.), the Instrumentarium Science Foundation and Academy of Finland #340539 (H.M.O.), Finnish Medical Foundation (S.S.), Younger Family and Bay Area Lyme Foundation (M.C.T.). The FinnGen project is funded by two grants from Business Finland (HUS 4685/31/2016 and UH 4386/31/2016) and the following industry partners: AbbVie Inc., AstraZeneca UK Ltd, Biogen MA Inc., Bristol Myers Squibb (and Celgene Corporation & Celgene International II Sàrl), Genentech Inc., Merck Sharp & Dohme LCC, Pfizer Inc., GlaxoSmithKline Intellectual Property Development Ltd., Sanofi US Services Inc., Maze Therapeutics Inc., Janssen Biotech Inc, Novartis AG, and Boehringer Ingelheim International GmbH. Following biobanks are acknowledged for delivering biobank samples to FinnGen: Auria Biobank (www.auria.fi/biopankki), THL Biobank (www.thl.fi/biobank), Helsinki Biobank (www.helsinginbiopankki.fi), Biobank Borealis of Northern Finland (https://www.ppshp.fi/Tutkimus-ja-opetus/Biopankki/Pages/Biobank-Borealis-briefly-in-English.aspx), Finnish Clinical Biobank Tampere (www.tays.fi/en-US/Research_and_development/Finnish_Clinical_Biobank_Tampere), Biobank of Eastern Finland (www.ita-suomenbiopankki.fi/en), Central Finland Biobank (www.ksshp.fi/fi-FI/Potilaalle/Biopankki), Finnish Red Cross Blood Service Biobank (www.veripalvelu.fi/verenluovutus/biopankkitoiminta) and Terveystalo Biobank (www.terveystalo.com/fi/Yritystietoa/Terveystalo-Biopankki/Biopankki/). All Finnish Biobanks are members of BBMRI.fi infrastructure (www.bbmri.fi). Finnish Biobank Cooperative -FINBB (https://finbb.fi/) is the coordinator of BBMRI-ERIC operations in Finland. The Finnish biobank data can be accessed through the Fingenious® services (https://site.fingenious.fi/en/) managed by FINBB. The work of the Estonian Genome Center, University of Tartu was funded by the European Union through Horizon 2020 research and innovation program under grants no. 810645 and 894987, through the European Regional Development Fund projects GENTRANSMED (2014-2020.4.01.15-0012), MOBEC008 and Estonian Research Council Grant PRG1291.

## Author contributions

Designed, conducted, and analyzed genetic and expression data: S.S., E.A., H.H., S.R., P.S., U.V., S.E.J., G.R.N., N.S.A., M.D., T.A., and H.M.O. Designed, conducted, and analyzed functional experiments: G.B., S.G., P.H., Q.F., B.T.L., and M.C.T. Mentorship and intellectual contributions: T.E., E.S., and I.L.W. Wrote the manuscript: H.M.O., S.S., E.A., M.C.T., Q.F., B.T.L., P.H., S.E.J., S.G., G.B., and T.A.

## Competing interests

S.S., H.M.O., M.C.T., G.B., S.G., P.H., N.S.A., and T.A. are co-inventors on a patent application 18/125,007, which is related to this work. The remaining authors declare no competing interests.

## Additional information

# FinnGen

Satu Strausz[1,2,3,4], Mark Daly[1] & Hanna M. Ollila[1,12,13,14] ✉

## Estonian Genome Centre

Erik Abner ⑩ [5,15], Urmo Võsa ⑩ [5], Sten Raak[5], Estonian Biobank Research Team & Tõnu Esko[5]

A full list of members and their affiliations appears in the Supplementary Information.

