## [Peer Review File · Nature Communications]

SCGB1D2 inhibits growth of *Borrelia burgdorferi* and affects susceptibility to Lyme diseaseREVIEWER COMMENTS

Reviewer #1 (Remarks to the Author):

The key finding of this manuscript is the identification of a single gene with variants that are linked to increased incidence of human Lyme disease. The increased incidence was identified in by GWAS using two human genetic data bases from European countries endemic for Lyme disease (Finland and Estonia). Thus, natural experimentation was used to identify the susceptibility allele in exposed populations. This is the first GWAS study to be published for human Lyme disease susceptibility, and is clearly novel and impactful. The identified gene, Secretoglobin family 1D member 2 (SCGB1D2) has not previously been associated with Lyme disease in humans or mice, highlighting the power of GWAS studies for unbiased identification of genetic association with disease. A missense variant predicted to effect structure was identified as likely responsible for the phenotype in the FinnGen GWAS, and the same variant was identified in the Estonia population as well. An MHC Class II allele was also linked, consistent with previous Class II linkage in patient studies.

The biological relevance of SNPs in SCGB1D2 was supported by its unique linkage to Lyme disease diagnosis and to association with patients hospitalized with spirochetal disease (majority Lyme disease). The identified missense variant was hypothesized to destabilize structure and modulate an unknown function of SCGB1D2.

A weakness of this study is the lack of robust assay/mechanism for the variant and failure to assess impact of this variant on related bacteria or other isolates of *B. burgdorferi*.

Specifically:

The absence of ascribed anti-bacterial function led to authors to design an assay to test variant SCGB1D2 alleles in vitro: the ability to function as a defensin was based on its production by sweat glands. The observed 15% inhibition of in vitro growth of *B. burgdorferi* is quite modest and difficult to determine if there is in vivo significance. Since there are differences in the ability of *B. burgdorferi* isolates to disseminate from the skin and cause systemic disease it seems pertinent to determine if there are different susceptibilities of *Borrelia* isolates to SCGB1D2 alleles.

Although other diseases were not associated with SCGB1D2 , it is possible that SCGB1D2 alleles might impact growth of other bacteria and formal inclusion/exclusion could be informative as to mechanism. Since SCGB1D2 appears to be bacteriostatic for *Borrelia* its role relative for other bacteria such as *E. coli* or *Leptospira* may be masked by other more potent defensins. An additional spirochete, *Leptospira* sps, may have low incidence in Finland and Estonia, allowing its susceptibility to be overlooked by the GWAS findings. Assessment with additional *B. burgdorferi* isolates and readily cultured *Leptospira* species could provide functional insight.

Understanding the biological relevance of SCGB1D2 would be enhanced by comparison of its physiological concentration in relevant tissues to the concentrations tested Fig 3. A comparative dose response with a second *Borrelia* strain or spirochetal strain might strengthen in vivo importance of the modest growth inhibition identified in vitro. Because calprotectin is another defensin previously demonstrated to significantly control of *B. Burgdorferi*, a dose response comparison could be illuminating.

The authors are urged to discuss two other gene families previously associated with greater severity or longer duration of Lyme disease in the US. The first is variants in TLR1 and TLR2 that have been reported to influence Lyme disease chronicity by the labs of Fikrig and Steere. In the second, Steere and colleagues have identified different alleles of MHC class II molecules in individuals with treatment refractory Lyme disease. Could the authors elaborate on the relative abundance of these Class II alleles in the European populations vs the US and speculate on whether differential representation could be impacting MIH linkage in the current work?

Reviewer #2 (Remarks to the Author):

Summary: The authors describe a genomic and functional study investigating host factors associating with susceptibility to *Borrelia burgdorferi*, the causative agent of Lyme disease. Using data from FinnGen, the authors identify 5,248 cases of Lyme disease and compare their genome-wide genetic data to >337,000 controls. They observe 2 regions that exceed the threshold of genome-wide significance; the HLA region and a gene-dense region on chromosome 1 encoding (among other genes) SCGB1D2, a secretoglobin protein family member primarily expressed in skin sweat glands. They provide additional evidence for the SCGB1D2 association by convincingly replicating it in an Estonian biobank. The lead variant is a missense variant resulting in a Pro > Leu change at position 53 of the protein. Functional studies indicate that the reference version of the protein significantly inhibits Bb growth compared to the variant, suggesting the variant is causal for increased Bb susceptibility. Overall this is an interesting study providing new evidence for host genetic variability impacting an infectious disease. The genomic analyses are capably run and the functional data support the genetic observation. I do have some comments/questions for consideration that I hope will improve the overall quality of the study.

Comments:

1) The analysis of the HLA region seems a bit superficial. As the authors correctly point out, the HLA region in general and HLA-DRB1*15:01 in specific are highly relevant to several infectious traits. Given the novelty of this association to Bb and the implications for immunity and vaccine design I feel a full fine-mapping analysis is warranted. Several tools are available for imputation of HLA alleles and amino acid variants, as well as frameworks for decomposition of the signal (see PMID: 34611364). I feel such an analysis could be easily performed and would add value to the manuscript as well as avenues for functional follow-up.

Along that line, I am also curious if the authors have an explanation for the relatively limited level of replication seen at the HLA locus in the EstBB cohort. Given the sample size and emphatic replication of the SCGB1D2 signal I'm surprised the statistical evidence isn't greater. A sentence or two putting this into some context would be informative.

2) I was also somewhat surprised that the authors only chose to replicate two loci in the EstBB sample, which contains roughly 3-fold more Bb cases. Why not perform a full meta-analysis to attempt to identify even more loci that associate with susceptibility? Seemingly this could be easily achieved with data on hand and wouldn't distract from the SCGB1D2 message.

3) The PheWAS data in FinnGen seems a little circular. Unless I'm mistaken, the same variant is identified in PheWAS to be associated with hospitalized spirochetal infection but with a largely overlapping sample set with the discovery cohort. So I'm not really sure what this is evidence of. It might be interesting to compare the effect of the SCGB1D2 variant in hospitalized vs unhospitalized Lyme disease patients to understand if the variant is more likely to mediate susceptibility or severity but the analysis stops short of this. I would consider a deeper analysis or re-framing the PheWAS portion to clarify the overall in samples.

4) Did the authors assess whether the rs2232950 variant is associated with SCGB1D2 expression in skin tissue? This would seem a natural analysis given the data on hand, and, since the biological hypothesis is that the variant protein reduces Bb inhibition it would be important to know the relative expression levels of this protein in people with and without the variant.

Minor comments:

Some minor typos and grammar errors occur throughout the manuscript detracting a bit from the overall presentation of the work.

The horizontal line denoting genome-wide significance mentioned in the figure 1 legend does not come through in the version I have. Please check.

Reviewer #3 (Remarks to the Author):

Strausz et al. present a very interesting and relevant study focusing on the identification of genetic factors for susceptibility to Lyme disease. Using the FinnGen database, they performed a GWAS and found a variant in the SCGB1D2 gene to be associated with increased risk. This could be independently replicated in the Estonian Biobank.

The authors found themselves in the happy situation that the lead variant was a missense variant with direct impact on protein function, and they followed up with an analysis of its functional impact and hypothesized role in *B. burgdorferi* growth.

Overall, the study is very well designed and executed, and the reading flow is great, also thanks to the straightforward results that allowed a functional follow-up, going beyond the mere reporting of a statistical association. Methods are appropriate, all results are credible, and both are presented in a coherent manner. I have little to criticize, apart from the following minor suggestions:

- It would be good to already mention the independent replication in EstBB in the abstract.
- HLA finemapping: Does the finemapped class II haplotype completely explain the top SNP signal? Adding DRB*15:01 and DQB1*06:02 as covariates in a conditional analysis would shed light on this question. Further, can this association also be replicated in EstBB? HLA associations related to antigen presentation might be strain specific, and so a replication (or non-replication) would be informative and could probably easily be done using the lead SNP if no HLA data is immediately available.
- Figure 2: It would be good to clarify in the figure legend that most hospitalized spirochete infection cases are indeed Lyme disease patients, and so the second phenotype is not an independent hit. This is described in the text, but it would be easier for the reader to not have to search for it, since it's an obvious question.
- Page 10, paragraph 2: If slight reduction suggest an active role of the gene in killing, it would be interesting to increase the concentration further, especially if the authors suggest to exploit it for drug development. I'm aware this would be a significant effort, and do not suggest this as a requirement for an acceptance of the paper.

Chris Hammer, Feb 6 2023

Point to point responses to reviewers
REVIEWER COMMENTS

Reviewer #1 (Remarks to the Author):

The key finding of this manuscript is the identification of a single gene with variants that are linked to increased incidence of human Lyme disease. The increased incidence was identified in by GWAS using two human genetic data bases from European countries endemic for Lyme disease (Finland and Estonia). Thus, natural experimentation was used to identify the susceptibility allele in exposed populations. This is the first GWAS study to be published for human Lyme disease susceptibility, and is clearly novel and impactful. The identified gene, Secretoglobin family 1D member 2 (SCGB1D2) has not previously been associated with Lyme disease in humans or mice, highlighting the power of GWAS studies for unbiased identification of genetic association with disease. A missense variant predicted to effect structure was identified as likely responsible for the phenotype in the FinnGen GWAS, and the same variant was identified in the Estonia population as well. An MHC Class II allele was also linked, consistent with previous Class II linkage in patient studies.

The biological relevance of SNPs in SCGB1D2 was supported by its unique linkage to Lyme disease diagnosis and to association with patients hospitalized with spirochetal disease (majority Lyme disease). The identified missense variant was hypothesized to destabilize structure and modulate an unknown function of SCGB1D2.

Comment 1: A weakness of this study is the lack of robust assay/mechanism for the variant and failure to assess impact of this variant on related bacteria or other isolates of *B. burgdorferi*.

Specifically:

The absence of ascribed anti-bacterial function led to authors to design an assay to test variant SCGB1D2 alleles in vitro: the ability to function as a defensin was based on its production by sweat glands. The observed 15% inhibition of in vitro growth of *B. burgdorferi* is quite modest and difficult to determine if there is in vivo significance. Since there are differences in the ability of *B. burgdorferi* isolates to disseminate from the skin and cause systemic disease it seems pertinent to determine if there are different susceptibilities of *Borrelia* isolates to SCGB1D2 alleles.

Although other diseases were not associated with SCGB1D2 , it is possible that SCGB1D2 alleles might impact growth of other bacteria and formal inclusion/exclusion could be informative as to mechanism. Since SCGB1D2 appears to be bacteriostatic for *Borrelia* its role relative for other bacteria such as *E coli* or *Leptospira* may be masked by other more potent defensins. An additional spirochete, *Leptospira* sps, may have low incidence in Finland and Estonia, allowing its susceptibility to be overlooked by the GWAS findings. Assessment with additional *B burgdorferi* isolates and readily cultured *Leptospiral* species could provide functional insight.

Understanding the biological relevance of SCGB1D2 would be enhanced by comparison of its physiological concentration in relevant tissues to the concentrations tested Fig 3. A comparative

dose response with a second *Borrelia* strain or spirochetal strain might strengthen in vivo importance of the modest growth inhibition identified in vitro. Because calprotectin is another defensin previously demonstrated to significantly control of *B. Burgdorferi*, a dose response comparison could be illuminating.

Response 1: We appreciate your comments and your suggestions have helped us strengthen the paper tremendously. To elucidate the function of SCGB1D2 on other *Borrelia burgdorferi* (*B. burgdorferi*) isolates, we obtained two separate *B. burgdorferi* isolates (ML23 and N40D10E9) genetically engineered with luciferase such that infection load could be measured in live animals over the course of an in vivo infection. We performed an in vivo experiment in mice where we examined SCGB1D2's impact on establishing infection. To ensure robustness of the experiment, we performed both the experiment and the analysis blinded, along with adding an additional control secretoglobin recombinant protein (SCGB3A1) from the same provider (Aviva Systems Biology) as a negative control. We incubated and co-infected the *B. burgdorferi* with 20 ug/mL SCGB1D2, or another secretoglobin, SCGB3A1 as a control, or performed the infection with *Borrelia* alone. While *Borrelia* without any protein, and the *Borrelia* incubated and co-infected with SCGB3A1 control protein established an infection, we found that the *Borrelia* incubated and co-infected with SCGB1D2 significantly impacted the kinetics of infection and were not able to establish infection in both ML23 and N40 isolates. This finding was extremely striking as seen in **Figure 4**.

Comment 2: The authors are urged to discuss two other gene families previously associated with greater severity or longer duration of Lyme disease in the US. The first is variants in TLR1 and TLR2 that have been reported to influence Lyme disease chronicity by the labs of Fikrig and Steere.

Response 2: Our new meta-analysis discovers a genome-wide association at the same TLR1 locus in chromosome 4 that has been examined in context of TLR responses and was earlier described in Lyme disease. The previously associated variant rs5743618 / 1805GG / Ser602Ile in TLR1 is genome-wide significant but similarly to earlier genome-wide scans at the region, not the most significant with the locus. Comparing LD between the lead variant in the meta-analysis (rs17616434, $P = 6.11 \times 10^{-11}$) and the previously described missense variant rs5743618 likely partially reflect the same signal in European populations ($r^2 = 0.78$) and in the Finnish population ($r^2 = 0.90$).

The TLR2 variant (Arg753Gln, rs5743708) did not associate with Lyme disease in this meta-analysis ($P=0.3357$, $\beta = 0.0224$, $se = 0.0232$).

We have added a section in the manuscript that describes the novel locus, its previous association with TLR responses and cites the earlier study where the missense variants have been examined in Lyme disease.

“We also observed an association within the Toll-like receptor 1 (TLR1) locus (rs17616434, $P = 6.11 \times 10^{-11}$). TLR1 in chromosome 4 and also TLR2 are interesting as coding variants at the loci have been previously associated with Lyme disease ¹. In our study we observed an association with TLR1 locus and the lead variant is a non-coding variant at 5’UTR of TLR1 and an eQTL for TLR1, TLR6 and TLR10 and associates robustly in FinnGen and in Estonian Biobank (Supplementary Figure 3). Additionally, it is in high LD with two missense variants at TLR1 (rs5743618, Ser602Ile, $r^2 = 0.90$; and rs4833095, Asn248Ser, $r^2 = 0.89$). The TLR1 locus has been reported as the main association for individual variation for TLR stimulation and response ². Furthermore, TLR1 SNPs including the missense variants have been associated with asthma or allergy and lymphocyte counts ³⁻⁵ and examined in infectious traits ^{6,7}, including Lyme disease ¹.”

Comment 3: In the second, Steere and colleagues have identified different alleles of MHC class II molecules in individuals with treatment refractory Lyme disease. Could the authors elaborate on the relative abundance of these Class II alleles in the European populations vs the US and speculate on whether differential representation could be impacting MIH linkage in the current work?

Response 3: This is an interesting question and we clarify the role of allele frequencies and abundance of the alleles in populations in the manuscript as follows:

“Furthermore, selection works relatively strongly at the HLA locus as infectious challenges can rapidly favour an HLA-allele that protects the population against a particular infection. In addition, the frequency of HLA alleles varies between populations and can affect the power to detect associations in different populations. Consequently, studies in Lyme disease will benefit from even larger genetic studies. Our findings provide yet another infectious disease trait that is associated with these same HLA alleles and raises the possibility that the same variants that contribute to infectious diseases also affect autoimmune and chronic disease traits in general.”

Reviewer #2 (Remarks to the Author):

Summary: The authors describe a genomic and functional study investigating host factors associating with susceptibility to *Borrelia burgdorferi*, the causative agent of Lyme disease. Using data from FinnGen, the authors identify 5,248 cases of Lyme disease and compare their genome-wide genetic data to >337,000 controls. They observe 2 regions that exceed the threshold of genome-wide significance; the HLA region and a gene-dense region on chromosome 1 encoding (among other genes) SCGB1D2, a secretoglobin protein family member primarily expressed in skin sweat glands. They provide additional evidence for the SCGB1D2 association by convincingly replicating it in an Estonian biobank. The lead variant is a missense variant resulting in a Pro > Leu change at position 53 of the protein. Functional studies indicate that the reference version of the protein significantly inhibits Bb growth compared to the variant, suggesting the variant is causal for increased Bb susceptibility. Overall

this is an interesting study providing new evidence for host genetic variability impacting an infectious disease. The genomic analyses are capably run and the functional data support the genetic observation. I do have some comments/questions for consideration that I hope will improve the overall quality of the study.

Comments:

Comment 1. The analysis of the HLA region seems a bit superficial. As the author correctly point out, the HLA region in general and HLA-DRB1*15:01 in specific are highly relevant to several infectious traits. Given the novelty of this association to Bb and the implications for immunity and vaccine design I feel a full fine-mapping analysis is warranted. Several tools are available for imputation of HLA alleles and amino acid variants, as well as frameworks for decomposition of the signal (see PMID: 34611364). I feel such an analysis could be easily performed and would add value to the manuscript as well as avenues for functional follow-up.

Along that line, I am also curious if the authors have an explanation for the relatively limited level of replication seen at the HLA locus in the EstBB cohort. Given the sample size and emphatic replication of the SCGB1D2 signal I'm surprised the statistical evidence isn't greater. A sentence or two putting this into some context would be informative.

Response 1. Thank you for the comment. We have performed extensive HLA analysis with population specific HLA sequencing and generated a population specific HLA imputation panel in Finland. We have now clarified and expanded these earlier analyses. Specifically, in FinnGen a Finnish cohort from the Finnish Red Cross was used to provide samples for HLA sequencing to build a reference panel for the Finnish population [PMID: 33575586] and a Pan-European reference panel for the Estonian population.

We then performed formal HLA fine mapping at variant level and saw strongest association with rs9276610. Logistic regression analysis at the HLA-allele level showed association with HLA-DQB1*06:02. Finally, HLA amino acids level showed association with amino acids that are part of HLA alleles that encode for HLA-DRB1*15:01 or DQB1*06:02 proteins:

We discovered leading associations with DRB1 position 0, an amino acid specific for DRB1*15:01 allele (beta = 0.17, se = 0.026, $P = 3.7 \times 10^{-11}$) and with DQB1 phenylalanine at position 9 that is found in HLA-DQB1*06:02 and in HLA-DQB1*04 alleles. We also expand this analysis in the Estonian biobank and provide SNP, HLA-allele and amino acid level associations as well as conditional analysis results in the main text and in the supplement.

Furthermore, we discuss the joint HLA effect driven by HLA-DRB1*15:01 and DQB1*06:02 and the differences across the two data sets as follows in the discussion:

“Furthermore, selection works relatively strongly at the HLA locus as infectious challenges can rapidly favour an HLA-allele that protects the population against a particular infection. In addition, the frequency of HLA alleles varies between populations and can affect the power to detect associations in different populations. Consequently, studies in Lyme disease will benefit

from even larger genetic studies. Our findings provide yet another infectious disease trait that is associated with these same HLA alleles and raises the possibility that the same variants that contribute to infectious diseases also affect autoimmune and chronic disease traits in general. ”

Comment 2. I was also somewhat surprised that the authors only chose to replicate two loci in the EstBB sample, which contains roughly 3-fold more Bb cases. Why not perform a full meta-analysis to attempt to identify even more loci that associate with susceptibility? Seemingly this could be easily achieved with data on hand and wouldn't distract from the SCGB1D2 message.

Response 2. This is a great suggestion. We have now obtained the full summary statistics from Estonian biobank and increased our sample including a later FinnGen release (release 10) which provided 412,181 individuals and provided 69,682 additional study participants from FinnGen. We then performed meta-analysis with Estonian biobank. The results show stronger association for SCGB1D2, for HLA, and finally discovers one additional locus at the TLR1 locus.

Comment 3. The PheWAS data in FinnGen seems a little circular. Unless I'm mistaken, the same variant is identified in PheWAS to be associated with hospitalized spirochetal infection but with a largely overlapping sample set with the discovery cohort. So I'm not really sure what this is evidence of. It might be interesting to compare the effect of the SCGB1D2 variant in hospitalized vs unhospitalized Lyme disease patients to understand if the variant is more likely to mediate susceptibility or severity but the analysis stops short of this. I would consider a deeper analysis or re-framing the PheWAS portion to clarify the overall in samples.

Response 3. The spirochetal infection phenotype is part of core endpoints of FinnGen which are provided by the project in each release. In the PheWAS analysis we examined all these core endpoints from FinnGen. To clarify the overlap of Lyme disease and hospitalized Lyme disease in the analysis we have modified the figure legend as follows:

“Figure 2. Phenome-wide associations of Secretoglobin 1D across disease traits. Volcano plot of Phenome-wide associations (PheWAS) from rs2232950 and 2,202 disease endpoints from FinnGen. Each point represents a trait. Vertical axis presents associated P-values at $-\log_{10}$ scale and the horizontal axis shows beta values. Other spirochetal diseases contain individuals who have been treated at hospital inpatient or outpatient clinics and are partially overlapping with Lyme disease as spirochetal diseases include individuals with Lyme disease.”

In addition, we computed the hospitalized versus non-hospitalized association statistics for SCGB1D2 but there was no evidence for SCGB1D2 affecting disease severity proxied by

hospitalization ($P = 0.23$) suggesting that the SCGB1D2 variant rs2232950 was related to infection and not hospitalization.

Comment 4. Did the authors assess whether the rs2232950 variant is associated with SCGB1D2 expression in skin tissue? This would seem a natural analysis given the data on hand, and, since the biological hypothesis is that the variant protein reduces Bb inhibition it would be important to know the relative expression levels of this protein in people with and without the variant.

Response 4. We have computed variant specific expression in skin from lower leg using data from GTEx. Rs2232950 did not associate with changes in relative expression of SCGB1D2 ($P = 0.67$).

Comment 5. Minor comments:

Some minor typos and grammar errors occur throughout the manuscript detracting a bit from the overall presentation of the work.

Response 5. We have gone carefully through the language and hope that it is now easier to read.

Comment 6. The horizontal line denoting genome-wide significance mentioned in the figure 1 legend does not come through in the version I have. Please check.

Response 6. We have corrected the genome-wide line.

Reviewer #3 (Remarks to the Author):

Strausz et al. present a very interesting and relevant study focusing on the identification of genetic factors for susceptibility to Lyme disease. Using the FinnGen database, they performed a GWAS and found a variant in the SCGB1D2 gene to be associated with increased risk. This could be independently replicated in the Estonian Biobank.

The authors found themselves in the happy situation that the lead variant was a missense variant with direct impact on protein function, and they followed up with an analysis of its functional impact and hypothesized role in *B. burgdorferi* growth.

Overall, the study is very well designed and executed, and the reading flow is great, also thanks to the straightforward results that allowed a functional follow-up, going beyond the mere reporting of a statistical association. Methods are appropriate, all results are credible, and both are presented in a coherent manner. I have little to criticize, apart from the following minor suggestions:

Comment 1. It would be good to already mention the independent replication in EstBB in the abstract.

Response 1. We obtained full summary statistics from the Estonian biobank and computed meta-analysis across the two cohorts. We now present results from the meta-analysis and discuss the reproducibility across the individual cohorts.

Comment 2. HLA finemapping: Does the finemapped class II haplotype completely explain the top SNP signal? Adding DRB*15:01 and DQB1*06:02 as covariates in a conditional analysis would shed light on this question. Further, can this association also be replicated in EstBB? HLA associations related to antigen presentation might be strain specific, and so a replication (or non-replication) would be informative and could probably easily be done using the lead SNP if no HLA data is immediately available.

Response 2. We performed conditional analysis adjusting for HLA-DQB1*06:02 allele and examined the association at the HLA locus. The P-value for the lead rsid in this conditioned analysis was $P = 0.02$. Overall these results suggest that the majority of the HLA signal is coming from signal that also marks HLA-DQB1*06:02.

*“Similarly, when adjusting the analysis for HLA-DQB1*06:02 we discovered that there was only weak residual signal remaining (rs9276610 P conditioned = 0.02) at the HLA locus suggesting that HLA-DQB1*06:02 explained the majority of the signal in this analysis although larger data sets might clarify the possible additional signals at the HLA locus.”*

Comment 3. Figure 2: It would be good to clarify in the figure legend that most hospitalized spirochete infection cases are indeed Lyme disease patients, and so the second phenotype is not an independent hit. This is described in the text, but it would be easier for the reader to not have to search for it, since it's an obvious question.

Response 3. We include this description in the figure legend as follows:

“Figure 2. Phenome-wide associations of Secretoglobin 1D across disease traits. Volcano plot of Phenome-wide associations (PheWAS) from rs2232950 and 2,202 disease endpoints from FinnGen. Each point represents a trait. Vertical axis presents associated P-values at $-\log_{10}$ scale and the horizontal axis shows beta values. Other spirochetal diseases contain individuals who have been treated at hospital inpatient or outpatient clinics and is partially overlapping with Lyme disease as spirochetal diseases include individuals with Lyme disease.”

Comment 4. Page 10, paragraph 2: If slight reduction suggest an active role of the gene in killing, it would be interesting to increase the concentration further, especially if the authors suggest to exploit it for drug development. I'm aware this would be a significant effort, and do not suggest this as a requirement for an acceptance of the paper.

Response 4. Thank you so much for the great suggestion. To estimate the possible active role of SCGB1D2 we tested the function of SCGB1D2 in an *in vivo* infection and increased the concentration of protein used to 20 ug/mL, higher than the 16 ug/mL tested *in vitro*. We obtained two separate *Borrelia burgdorferi* (Bb) isolates genetically engineered with luciferase (ML23 and N40) such that infection load could be measured in live animals over the course of an *in vivo* infection. We performed an *in vivo* experiment in mice where we examined SCGB1D2's impact on establishing infection over the course of 10 days (Figure 4 and supplemental figure 14). Here, we show the trajectory of infection using Bb engineered to incorporate luciferase and measuring total flux of light emitted 15 minutes after luciferin injection. The mice were imaged with IVIS for 1 minute exposure for total flux signal over the host body. The images are shown with intensity normalized to day zero immediately post infection by measuring luciferin for 15 minutes. Initially the Bb (both ML23 and N40 strains) localized to the site of infection on the dorsal side of the prone animal. After which, spirochetes disseminate throughout the host, as indicated by the total flux signal increasingly widening on days 3, 7, and 10. To ensure robustness of the experiment, we performed both the experiment and the analysis blinded, along with an additional control secretoglobin recombinant protein (SCGB3A1) from the same provider as a negative control. We incubated and co-infected the *Borrelia* with 20 ug/mL SCGB1D2, or SCGB3A1 as a control, or performed the infection with *Borrelia* alone. While *Borrelia* without any additional added protein, and the *Borrelia* incubated and co-infected with SCGB3A1 control protein established an infection, we found that the *Borrelia* incubated and co-infected with SCGB1D2 significantly impacted the kinetics of infection and were not able to establish infection in both ML23 and N40 isolates. This finding was extremely striking as seen in Figure 4. What is most clear is that the Bb injected along with SCGB1D2 is not able to grow enough to establish an infection, and further experiments investigating the direct impact of SCGB1D2 on the bacteria while beyond the scope of this paper, would provide very important mechanistic information that we would like to investigate further in future studies.

References:

- 1 Strle, K., Shin, J. J., Glickstein, L. J. & Steere, A. C. Association of a Toll-like receptor 1 polymorphism with heightened Th1 inflammatory responses and antibiotic-refractory Lyme arthritis. *Arthritis Rheum* **64**, 1497-1507, doi:10.1002/art.34383 (2012).
- 2 Mikacenic, C., Reiner, A. P., Holden, T. D., Nickerson, D. A. & Wurfel, M. M. Variation in the TLR10/TLR1/TLR6 locus is the major genetic determinant of interindividual difference in TLR1/2-mediated responses. *Genes Immun* **14**, 52-57, doi:10.1038/gene.2012.53 (2013).
- 3 Donertas, H. M., Fabian, D. K., Valenzuela, M. F., Partridge, L. & Thornton, J. M. Common genetic associations between age-related diseases. *Nat Aging* **1**, 400-412, doi:10.1038/s43587-021-00051-5 (2021).
- 4 Chen, M. H. *et al.* Trans-ethnic and Ancestry-Specific Blood-Cell Genetics in 746,667 Individuals from 5 Global Populations. *Cell* **182**, 1198-1213 e1114, doi:10.1016/j.cell.2020.06.045 (2020).
- 5 Ferreira, M. A. *et al.* Shared genetic origin of asthma, hay fever and eczema elucidates allergic disease biology. *Nat Genet* **49**, 1752-1757, doi:10.1038/ng.3985 (2017).
- 6 Masin, P. S. *et al.* Genetic polymorphisms of toll-like receptors in leprosy patients from southern Brazil. *Front Genet* **13**, 952219, doi:10.3389/fgene.2022.952219 (2022).

- 7 Qi, H. *et al.* Toll-like receptor 1(TLR1) Gene SNP rs5743618 is associated with increased risk for tuberculosis in Han Chinese children. *Tuberculosis (Edinb)* **95**, 197-203, doi:10.1016/j.tube.2014.12.001 (2015).

REVIEWERS' COMMENTS

Reviewer #2 (Remarks to the Author):

The authors have adequately addressed my comments. I have no further concerns and support publication.

Reviewer #3 (Remarks to the Author):

The authors have addressed all the comments and suggestions I previously raised in a satisfactory and comprehensive manner. The revisions have significantly improved the clarity and quality of the manuscript, and I believe it is now ready for publication. In light of these improvements, I have no further concerns or suggestions for revisions.